# Regulation of Mesothelial Cell Fate during Development and Human Diseases

**DOI:** 10.3390/ijms231911960

**Published:** 2022-10-08

**Authors:** Toshiaki Taniguchi, Hiroyuki Tomita, Tomohiro Kanayama, Kazumasa Mogi, Yoshihiro Koya, Yoshihiko Yamakita, Masato Yoshihara, Hiroaki Kajiyama, Akira Hara

**Affiliations:** 1Department of Tumor Pathology, Gifu University Graduate School of Medicine, Gifu 501-1194, Japan; 2Department of Obstetrics and Gynecology, Graduate School of Medicine, Nagoya University, Nagoya 466-8560, Japan; 3Bell Research Center, Department of Obstetrics and Gynecology Collaborative Research, Graduate School of Medicine, Nagoya University, Nagoya 466-8560, Japan

**Keywords:** mesothelial cell, *Wt1*, differentiation, lineage tracing, cancer-associated fibroblast

## Abstract

Mesothelial cells (MCs) play a classic role in maintaining homeostasis in pleural, peritoneal, and pericardial cavities. MCs work as lubricants to reduce friction between organs, as regulators of fluid transport, and as regulators of defense mechanisms in inflammation. MCs can differentiate into various cells, exhibiting epithelial and mesenchymal characteristics. MCs have a high potential for differentiation during the embryonic period when tissue development is active, and this potential decreases through adulthood. The expression of the Wilms’ tumor suppressor gene (*Wt1*), one of the MC markers, decreased uniformly and significantly from the embryonic period to adulthood, suggesting that it plays a major role in the differentiation potential of MCs. *Wt1* deletion from the embryonic period results in embryonic lethality in mice, and even *Wt1* knockout in adulthood leads to death with rapid organ atrophy. These findings suggest that MCs expressing *Wt1* have high differentiation potential and contribute to the formation and maintenance of various tissues from the embryonic period to adulthood. Because of these properties, MCs dynamically transform their characteristics in the tumor microenvironment as cancer-associated MCs. This review focuses on the relationship between the differentiation potential of MCs and *Wt1*, including recent reports using lineage tracing using the Cre-loxP system.

## 1. Introduction

Mesothelial cells (MCs) form the surface layer of serosa-like paving stones, including the peritoneum, pleura, and pericardium, with a cell diameter of ~25 μm [1]. In addition, the surface of visceral organs, such as the lung, heart, liver, intestine, colon, uterus, and tunica vaginalis testis, is lined with MCs. So far, there is much evidence of MC differentiation from the mesoderm during the embryo stage. However, in the adult stage, the capacity and capability that MCs have are confusing.

Many genes expressed in MCs have been reported in the previous literature. Among these genes, the Wilms’ tumor suppressor gene (*Wt1*) is expressed in nearly all MCs [2]. Thus, *Wt1*-expressing MCs (*Wt1*^+^ MCs) have multipotent progenitor cells that can transdifferentiate into epithelial and mesenchymal cells. In the inflammatory condition, hepatic stellate cells (HSCs) and pancreatic stellate cells (PSCs) are partially derived from MCs in adult mice [3,4,5,6]. Although the capacity is limited in the adult stage compared to the embryo stage, adult MCs are likely to transdifferentiate other cell types in specific conditions (i.e., inflammation or tumorigenesis in somatic body cavities).

This study reviewed the differences between embryonic and adult stages in the capacity of MC differentiation, particularly related to *Wt1*.

## 2. MCs and *Wt1* Gene

### 2.1. General Information on MCs

The roles of MCs include reducing friction between organs as a lubricant, regulating fluid transport as a semipermeable membrane, and regulating immune function [7]. MCs produce hyaluronic acid and sialomucin, which reduce the coefficient of kinetic friction, contribute to lubrication between organs, and protect injured tissues [8,9]. Tight junctions involved in cell–cell junctions are important for developing cell-surface polarity and maintaining a semipermeable diffusion barrier, which passively transports fluid across the peritoneum [10]. Furthermore, MCs trigger inflammatory responses to bacteria and viruses, induce and activate immune cells, and participate in tissue repair after inflammation by regulating coagulation and preventing reinvasion from damaged areas [11].

On the structure of MCs, they form specific glycocalyx (a complex composed of glycolipids, proteoglycans, and glycosaminoglycans that coat the cell surface) to maintain homeostasis in the body cavity, and most glycosaminoglycans in the glycocalyx are composed of the hyaluronan family [12,13,14]. Hyaluronic acid (HA) is composed of repeating D-glucuronic acid and N-acetyl-D-glucosamine, which is synthesized by uridine diphosphate glucose dehydrogenase and HA synthase (HAS) [15]. HAS has three major isozymes, HAS I, HAS II, and HAS III, in which HAS II is necessary for survival [16]. The role of hyaluronan depends on the molecular weight, with high-molecular-weight synthesis mainly driven by HAS II and low-molecular-weight synthesis particularly by HAS III. Even so, the mechanism of its regulation remains unclear [17]. Further studies are needed to clarify the role of HA in the developmental and pathological processes of MCs. The glycocalyx formed by MCs lubricates and protects the serosal surface from frictional damage arising from the movement of organs and other surfaces. However, the non-cross-linked organization of the mesothelial glycocalyx is poor in the recognition of, or as a physical barrier against, bacteria and viruses. Therefore, MCs display multiple pattern-recognition receptors that recognize carbohydrates and lipopolysaccharides on the surface of microbial pathogens, such as bacteria and viruses, to release inflammatory mediators and initiate inflammation [12,18]. MCs lead to the activation of macrophages mobilized from mesothelial subcutaneous tissue by CSF1 secretion [19], express class II major histocompatibility complex molecules, and, therefore, modulate lymphocytes [20].

### 2.2. Wt1 in MC Differentiation

MCs derived from the embryonic mesoderm exhibit characteristics such as epithelial cells forming tight and gap junctions and desmosomes, while exhibiting mesenchymal features expressing vimentin and N-cadherin [1,21,22]. *Wt1* may be adjusting for these characteristics [23]. The *Wt1* gene maps to chromosome 11p13, which was first identified in 1990 as a strong candidate predisposition gene for the Wilms’ tumor [24]. *Wt1* is widely expressed in mesodermal progenitors during embryogenesis, and the loss of *Wt1* in mice lacks kidney, adrenal, gland, gonad, spleen, and coronary vasculature [25,26,27]. In the kidneys, nephron progenitor cells derived from the intermediate mesoderm led to underdeveloped kidneys which reduced the expression of Wnt4, a regulator of the mesenchymal-to-epithelial transition to nephrons due to the loss of *Wt1* [28,29]. In contrast, in the heart, epicardial MCs contribute to the formation of coronary vessels through epithelial-to-mesenchymal transition (EMT), and the loss of *Wt1* leads to underdeveloped coronary vasculature with no effect on epicardial formation [30,31,32].

Although *Wt1* is widely expressed during embryogenesis, *Wt1* expression in adult mice is restricted to a small percentage of cells, such as the podocyte cells of the kidney and gonads, ~1% of cells in the bone marrow, and MCs [23,33]. Nevertheless, adult mice with induced knockout of *Wt1* have undeveloped glomerulosclerosis, spleen and exocrine pancreatic atrophy, loss of bone and fat mass, and defects in red blood cell formation [23,34]. These findings suggest that *Wt1*-expressing cells have stem-cell-like properties and maintain homeostasis in adulthood. Today, it is possible to trace the lineage of specific cells by generating conditional knockout experimental animals with the Cre-loxP system. As a result, there is an increasing number of reports on the lineage tracing of *Wt1*^+^ MCs with respect to their differentiation and function during the embryonic period and adulthood.

## 3. Differentiation of MCs into Fibroblasts

### 3.1. Differentiation of MCs into Fibroblasts Related to Peritoneal Dialysis (PD)

PD is an effective and affordable renal replacement therapy, resulting in less frequent hospital visits and significantly improved quality of life compared to hemodialysis [18]. However, it is used by only ~11% of the total global dialysis population [35]. One cause is nonphysiological PD solutions that are bioincompatible for sustained peritoneal exposure and provoke MC injury and peritoneal inflammation [36,37]. In a study comparing the thickness of the submesothelial compact collagenous zone in patients undergoing PD, PD-induced fibrosis of the peritoneum leads to reduced dialysis efficiency [38]. Normal peritoneal fibroblasts are scattered in the submesothelial connective tissue and express neither myofibroblastic nor MC markers. However, many fibroblasts associated with PD showed a myofibroblast phenotype expressing α-smooth muscle actin (α-SMA) and accompanied by the expression of cytokeratins suggestive of MC [39,40].

MCs that transition to myofibroblasts upregulate the transcription factors of Snail, ZEB, and Twist. *Wt1* promotes EMT by directly activating the Snai1 promoter and directly repressing the *Cdh1* (*E-cadherin*) promoter during development in the mesenchymal progenitor cells [30]. Transforming growth factor-β (TGF-β) has been reported as a representative and important mediator in inducing EMT [41,42,43]. Normal MCs homeostatically produce factors, such as bone morphogenetic protein, that counteract the induction of EMT and inhibit TGF-β expression [44,45]. In vitro and in vivo, PD fluids with a high glucose degradation product (GDP) concentration induce TGF-β production and EMT in MCs [46,47]. MC stimulation by TGF-β1 and interleukin (IL)-1β activates TGF-β-activated kinase 1, increases the expression of extracellular signal-regulated kinase 1/2, nuclear factor-κB, and Snail, and promotes EMT [48,49,50]. Hepatocyte growth factor, BMP7, vitamin D analogs, and corticosteroids inhibit EMT and peritoneal fibrosis [44,45,51,52]. In a study that used *Wt1*-dependent CreER-expressing mice for the lineage tracing of *Wt1*^+^ MCs, the intraperitoneal injection of hypochlorite or dialysis solution, containing 4.25% glucose and 40 mM GDP, induced peritoneal fibrosis, and *Wt1*^+^ MCs underwent tissue repair through cell cycle upregulation, as analyzed by increased Ki-67 expression. In contrast, within the submesothelial scar, ~15.9% and 16.5% of the cells differentiated into myofibroblasts expressing α-SMA [53]. In addition, myofibroblasts derived from *Wt1*^+^ MC express platelet-derived growth factor receptor-β (PDGFR-β), and the PDGFR tyrosine kinase inhibitor imatinib significantly attenuated the accumulation of α-SMA myofibroblasts and reduced the fibrotic thickening of the peritoneum.

### 3.2. Differentiation of Wt1^+^ MCs into Fibroblasts

*Wt1*^+^ MCs in the pancreas are important for the formation and maintenance of PSCs, the fibroblasts of the pancreas. PSCs are usually located around the acinar cells, ducts, and vessels, and comprise ~4% of the total pancreas, and quiescent PSCs are necessary for pancreatic exocrine stability and regeneration [54,55,56]. In a study that used *Wt1*-dependent CreER-expressing mice for the lineage tracing of *Wt1*^+^ MCs, E9.5 and E15.5 showed epithelial features of MCs, whereas E10.5 to E14.5 showed a puncture pattern characteristic of mesenchymal cells in the submesothelium and EMT features [57]. The induction of *Wt1* conditional knockout in MCs between E9.5 and E12.5 showed a significant delay in the development of ventral pancreatic buds at E16.5, characterized by fewer and less dense gland tufts compared to controls. In adult mice, *Wt1*^+^ cells are confined to MCs only and do not contribute to PSCs. During *Wt1* conditional knockout in adult mice, glandular structures show reduced intercellular adhesion, a rounded shape, and disrupted exocrine tissue structure [58]. Caerulein-induced pancreatitis in these mice partially rescues glandular tissue by causing de novo *Wt1* expression in PSCs, even when *Wt1* is deleted from MCs.

HSCs, the fibroblasts of the liver, are also thought to be derived from *Wt1*^+^ MCs. A study that used *Wt1*-dependent CreER-expressing mice for the lineage tracing of *Wt1*^+^ MCs supports the contribution of MCs to HSCs and perivascular mesenchymal cells at E11.5 to E12.5. Liver or biliary injury by carbon tetrachloride (CCl_4_) or bile duct ligation in mice also results in the differentiation of *Wt1*^+^ MCs into HSCs or myofibroblasts. Still, the antagonization of TGF-β1 and TGF-β3 by soluble TGF-β receptor 2 decreases the differentiation of *Wt1*+ MCs [59]. Previous studies using collagen-driven Cre or *Wt1*-Cre to track myofibroblasts all tracked collagen-producing myofibroblasts and showed differentiation as part of their composition, but their role remains unclear [60,61]. Recent studies have shown that activated HSCs derived from *Wt1*^+^ MCs act to suppress classical myofibroblasts involved in fibrogenesis [62]. In this study, to define the subpopulation of active HSCs, *Wt1* expression was separated into high-, intermediate-, and negative-expressing cells and cultured. Only high-expressing cells showed a rounded morphology that does not exhibit classic myofibroblast characteristics, and transcriptome data showed a high incidence of mesothelium-related profiles. *Wt1* deletion reduced the mean circularity of these cells by 17.5% and was accompanied by a 67.56% increase in fibrosis due to CCl4 injury, indicating that HSCs derived from *Wt1*^+^ MCs are inhibitory to fibrosis.

### 3.3. Differentiation of MCs into Myofibroblasts in Cancer

Peritoneal dissemination is the primary metastatic route for gastric, ovarian, and pancreatic cancers, and contact is necessarily made with MCs during seeding formation in other organs. MCs have been thought to act as a protective barrier against cancer progression [63,64]. Cancer with peritoneal dissemination often results in ascites effusions that contain numerous cytokines, including TGF-β, which differentiate MCs into the cancer-associated fibroblast (CAF) phenotype, suggesting their involvement in the tumor microenvironment [65,66,67,68]. Integrin α5 is involved in the enhanced adhesion of ovarian cancer in peritoneal dissemination and is highly expressed in single cells in ascites fluid or spheroids formed from multiple cells [69,70]. Recent reports have shown that CAFs in spheroids that receive TGF-β signaling from ovarian cancer secrete epidermal growth factor and induce integrin α5 expression in ovarian cancer, promoting adhesion to the peritoneum [71]. PDGF signaling is important for CAF survival [72], and imatinib reduces CAF activity and leads to a rapid decrease in peritoneal adhesion [71]. MCs differentiated into CAF-like phenotype by TGF-β stimulation acquire platinum resistance by upregulating FN1 expression and activating Akt1 signaling in ovarian cancer cells [73]. MCs may also be involved in angiogenesis within tumors [74].

Malignant ascites contain high vascular endothelial growth factor (VEGF) concentrations [66,75], a source not only of cancer but also of MCs [76]. Normal MCs also produce VEGF homeostatically, and MC stimulation with TGF-β, IL-1β, and fibroblast growth factor 2 increases VEGF secretion, suggesting tumor angiogenesis [74,77,78,79]. Bevacizumab, a monoclonal antibody targeting all VEGF-A isoforms, is also useful in platinum-resistant ovarian cancer, suggesting a strong involvement of MC in the tumor [80,81]. These reports support the need to target MC in the tumor microenvironment as a novel therapeutic strategy.

A novel strategy for preventing peritoneal dissemination has recently been reported, focusing on the ability of vitamin D to recover MC function by inhibiting TGF-β-stimulated MC mesenchymal transition secreted from ovarian cancer cells. In this study, vitamin D recovers MC function and significantly reduces the peritoneal dissemination of ovarian cancer by suppressing thrombospondin-1 expression [82]. A study using mice spontaneously developing pancreatic cancer found a population of podoplanin-positive CAFs in the stroma that is suggested to be derived from MCs [4]. Subpopulations involved in the poor response to anti-PD-L1 therapy have been identified by profiling the population with single-cell RNA sequencing (RNAseq). A recent study that used *Wt1*-dependent CreER-expressing mice for the lineage tracing of *Wt1*^+^ MCs identified MC-derived CAFs in the cancer stroma. This study found antigen-presenting CAFs from MCs that induce regulatory T cells and contribute to immune evasion [83].

## 4. Contribution of MCs to Blood Vessels

### 4.1. Wt1^+^ MCs in Heart Development

*Wt1* is highly expressed in epicardial MCs during fetal heart development and is downregulated as epicardial EMT progresses. Some MCs undergoing EMT after E12.5 in mice form epicardial-derived progenitor cells (EPDC) [84]. EPDCs migrate into the myocardium and differentiate into stromal fibroblasts and smooth muscle cells of the coronary vasculature [85,86]. *Wt1* deletion during the embryonic period in mice causes death by E14.5 with pericardial effusion at E13.5 [87]. In the mice model where the *Wt1* gene was conditionally knocked out by Gata5-Cre, a decrease in mesenchymal progenitor cells and their derivatives resulted in coronary artery dysplasia and death by E16.5 due to the accumulation of edema and blood in systemic veins [30]. A study that used *Wt1*-dependent CreER-expressing mice for the lineage tracing of *Wt1*^+^ MCs in the epicardium has shown that they form part of the coronary arteries in neonatal and adult stages [88]. In addition, lineage tracing data of *Wt1*^+^ MCs in the epicardium in adulthood have confirmed not only the expression of α-SMA, a vascular smooth muscle marker, but also CD31, a vascular endothelial cell marker. However, it is not possible to exclude that *Wt1*-expressing endothelial cells may have contributed to these labeled vascular cells.

*Wt1* expression in the heart is not restricted to the epicardium but is also present in the developing myocardial layer after E12.5 in mice and at 5 weeks postfertilization in humans [89]. In human and mouse myocardial layers, *Wt1* expression is observed in the endothelial cells of small capillaries and larger coronary vessels, and this decreases from the prenatal stage to adulthood [90,91]. In mice with myocardial infarction, *Wt1* expression is upregulated in endothelial cells in the infarct zone and border zone of the heart [90]. Over time, vessels in the infarcted region mature and form a fibrotic scar, and *Wt1* expression disappears. Endogenous *Wt1* expression in cardiac endothelial cells prevents the use of *Wt1*-Cre for the cell lineage tracing of epicardial MCs to coronary endothelial cells [92,93]. A study using Apln-CreER mice engineered to lineage trace subepicardial endothelial cells has shown that subepicardial endothelial cells contribute to the formation of coronary endothelial cells during the process of cardiogenesis [94].

### 4.2. Role of Wt1 in MC Differentiation Related to Tumor Angiogenesis

The process of angiogenesis in the tumor microenvironment is considered one of the key features of cancer [95,96]. Tumor and bone-marrow-derived cells, and other stromal cells, release paracrine VEGF, which increases vascular branching and promotes tumor vascular development [97]. The combination of conventional chemotherapy and the VEGF inhibitor bevacizumab has increased the survival rates of patients with advanced colorectal and lung cancers [98,99].

Pericytes are specialized mesenchymal cells with finger-like processes that wrap around the endothelial tubes of blood vessels [96]. PDGFR-β-positive pericytes are recruited to the periphery of endothelial cells by PDGF-B released from endothelial cells, tumors, and platelets, providing mechanical and physiological support for the vessel. Experiments using PDGFR inhibitors, in combination with VEGFR inhibitors, prevent islet cancer growth by inducing pericyte detachment and tumor vascularity disruption in mice with end-stage islet cancer [100]. In contrast, using anti-VEGF drugs in mice deficient in pericytes does not affect antitumor efficacy [101]. In addition, a study using mice with conditional knockout of PDGF-B in platelets has shown that the impaired pericyte coverage of tumor blood vessels promotes metastasis and, thus, the need for vascular integrity maintenance [102].

High-level *Wt1*-expressing tumors correlate with promoted angiogenesis, demonstrating that *Wt1* regulates VEGF expression [103,104]. In a study using the Ewing’s sarcoma cell line, *Wt1*-expressing tumors increased the expression of antiangiogenesis-promoting molecules, such as VEGF, MMP9, Ang-1, and Tie-2 [105]. In addition, in vivo experiments have shown that *Wt1* deletion in tumors significantly suppresses blood vessel and tumor formation.

Previously, it was found that *Wt1* is upregulated in endothelial cells of various tumors [106,107]. Mice with endothelial-cell-specific *Wt1* knockout using Tie2-CreER or VE-cadherin-CreER have reduced vascular density and tumor volume in melanoma and lung cancer [108]. In this study, mice with the *Wt1*-CreER genotype showed that ~50% of tumor stromal cells undergo *Wt1*-Cre-mediated recombination. In addition, *Wt1* coexpression with markers of endothelial cells, hematopoietic progenitor cells, myeloid cells, and pericytes has been observed in human tumor samples, supporting a high *Wt1* contribution to the tumor stroma.

## 5. MCs and Adipocytes

Adipose tissue is mainly classified into white adipose tissue (WAT) and brown adipose tissue (BAT) [109]. WAT further exists as subcutaneous and visceral WAT. Subcutaneous WAT is located under the skin and accounts for the highest percentage of adipose tissue [110]. Visceral WAT surrounds the perirenal, gonad, epicardium, retroperitoneum, cartilage, and mesentery [111]. WAT consists mainly of white fat cells with monoblastic fat droplets, which store excess energy in fat droplets as triglycerides and decompose them into glycerol and free fatty acids as necessary to resupply them throughout the body as an energy source [112]. BAT exists in locally limited quantities composed mainly of brown fat cells with the morphological feature of having small fat droplets that are multifocal, and numerous mitochondria are present around the fat droplets [113]. Uncoupling protein 1 (UCP1) on the inner mitochondrial membrane of brown adipocytes has the function of deconjugating oxidative phosphorylation in mitochondria, eliminating the proton concentration gradient without involving ATP synthesis, dissipating it as heat energy, and contributing to high heat production [114].

*Wt1* expression in adipose tissue is restricted to visceral WAT and is not detectable in subcutaneous WAT and BAT [115]. *Wt1* represses the BAT gene signature, and thermogenic genes, such as Ucp1 and Prdm16, are expressed in visceral WAT in mice with adipocyte-specific *Wt1* deletion [116]. It is presumed that *Wt1* leads to increased WAT and decreased BAT, and is associated with obesity, diabetes, and cardiovascular disease due to metabolic dysfunction. In general, the accumulation of subcutaneous WAT has no significant effect on mortality risk, but the accumulation of visceral WAT with *Wt1* expression leads to increased mortality risk [117]. BAT can be readily induced in subcutaneous WAT, but visceral WAT expressing *Wt1* is resistant to induction [118]. A study using heterozygous *Wt1* knockout mice has shown that the expression of thermogenic genes, such as Ucp1, is enhanced in visceral WAT [119]. Retroviral *Wt1* expression in brown preadipocytes also reduces the expression of thermogenic genes, such as Ucp1, during in vitro differentiation. These findings suggest that reducing *Wt1* expression in visceral WAT, a more intra-abdominal fat reservoir, may improve the percentage of brown preadipocytes and be a novel therapeutic strategy in metabolic diseases.

In a study of lineage tracing in adipose tissue, using mice expressing CreER in a *Wt1*-dependent manner, induction with tamoxifen administration at E14.5 and tissue analysis at 1.2 years of age showed that the epididymal fat (77%), epicardial (66%), visceral (47%), and mesenteric (28%) contributions of *Wt1*^+^ cells are confirmed in WAT. In contrast, no positive adipocytes derived from *Wt1*-expressing cells are present in subcutaneous WAT and BAT [120]. In addition, adipose progenitor cell-surface markers are expressed in *Wt1*-positive cells in adipose tissue in adulthood. Thus, the role of *Wt1*^+^ MCs as a source of visceral progenitor adipocytes during embryonic and adulthood is widely accepted, and subsequent studies have used *Wt1* for lineage tracing analysis [121]. In a recent study, MCs have been reported not to be a source of adipocytes in adults [122]. In this study, RNAseq of WAT in the epididymis identifies MC populations and progenitor adipocyte populations, and identifies that not only MC populations but also progenitor adipocyte populations express *Wt1*. In a previous study, single-cell RNAseq of WAT confirmed *Wt1* expression in the non-MC stromal population [123,124]. They also identified keratin 19 (Krt19) as a more MC-specific marker in the process of single-cell RNAseq [122]. Interestingly, data from lineage tracing analysis with *Wt1* and Krt19, respectively, found no involvement of Krt19^+^ cells in adipocytes. These data show that *Wt1* expressed by progenitor adipocytes should be distinguished from *Wt1*^+^ MCs. In visceral WAT, *Wt1* is detected in the stromal vascular fraction (e.g., endothelial cells, immune cells, and progenitor adipocytes). There, it differentiates not only into adipocytes but also into muscle cells and osteoblasts in vitro, although their role during adulthood remains unclear [115]. In tumor microenvironments, such as breast and pancreatic cancers, adipocytes differentiate into fibroblast-like cells by Wnt signaling and promote tumor development [125,126,127]. These are currently unknown in terms of their association with *Wt1*, which is interesting.

## 6. Conclusions

Finally, we summarized the capacity of MC differentiation during the embryonic and adult stages (Figure 1).

The control of MC, especially when expressing *Wt1* differentiation, is an important issue in the treatment of human diseases, such as inflammation, cancer, and injury, as well as PD. The recovery or modification of MC function may be one of the potential novel therapeutic strategies for these diseases and situations. Although the availability of MC control is still unknown, MC definitely has the potential for therapeutic development, and research on MCs needs to be further accelerated.

## Figures and Tables

**Figure 1 ijms-23-11960-f001:**
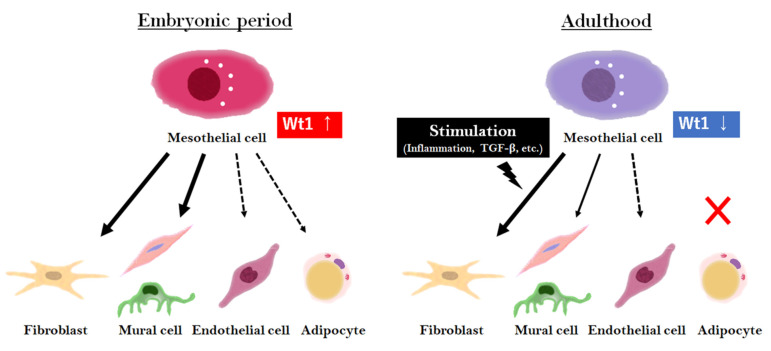
Overview of MC differentiation in the embryonic and adult stages. In the embryonic period, MCs highly express *Wt1*, contributing to various tissue formations. In adulthood, *Wt1* expression in MCs is low and limited to a few populations, but is involved in tissue repair with differentiation upon stimulation, such as inflammation or injury.

## Data Availability

Not applicable.

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
