# Peer review of "Regulation of Mesothelial Cell Fate during Development and Human Diseases"

_ijms, 2022, doi:10.3390/ijms231911960_

Round 1
Reviewer 1 Report
In the review written by Tonigashi and colleagues entitled "Regulation of mesothelial cell fate during development and hu- man diseases," the role of mesothelial cells in tissues, organs with a look at their role in the tumor microenvironment is described. The review is of interest and addresses a topic not often covered such as the role of mesothelial cells. It is well written and I find no substantial suggestions to make. Therefore, I consider it publishable in the present format.
Author Response
Dear Editor and Reviewers #1 and #2:
We appreciate your positive comments on our manuscript. Thank you for taking the time to provide detailed suggestions that have significantly all your comments. Our manuscript has improved our manuscript. as a result of your insightful comments.
Point-by-point responses to all the comments of Reviewer #2 have been provided below.
Thank you.
Hiroyuki Tomita
-------------------------------------------------------------------------------------------
Response to Reviewer #2:
Reviewer #2 Comments to the Author;
Generally, The aspects of WT1 in regulating mesothelial cell Plasticity are interesting topics to be investigated in developmental and pathological processes, however, this review is too descriptive and lack the molecular mechanism in detail. The transgenic animal models or human genetic association relevances did not present in the review to justify how WT1 regulate the mesothelial cell plasticity during the developmental and pathological stages.
Response:
We thank you for your efforts in reviewing our manuscript. Please find below our point-by-point responses to all your comments.
1) The detail molecular mechanism underlying how does MS regulate surface or secreted the clycocalyx complex, such as HA, glycosaminoglycans and sialomucine proteins respond to extracellular matrix remodeling during the developmental and pathological processes.
Response:
Thank you for your comment. The function of hyaluronic acid in mesothelial cells varies with molecular weight size. However, studies tracing the adjustment mechanism are currently poor. One of our future research topics is the effect of hyaluronic acid synthesis in mesothelial cells on the developmental process of cancer. Therefore, we have added a simplified description of the synthesis and role of hyaluronic acid in the revised manuscript. We have added several sentences to the revised manuscript on this topic (page 2, lines 15–23).
2) The integrated logical informations linked to WT1 regulating chromosomal architecture, epigenomic control or promoters activity between fibroblast, myofibroblast and SMC differentiations respond to extra cellular matrix cue during the different developmental or pathological stages needed to be addressed in more detail.
Response:
Thank you for your insightful suggestion. The epithelial–mesenchymal transition of epicardial mesothelial cells during development is a necessary process for forming coronary arteries. In addition to directly activating the Snai1 promoter, a regulator of epithelial–mesenchymal transition, in the process, Wt1 also directly represses E-cadherin expression. Therefore, we have added the above statement to the revised manuscript (page 3, lines 18–20).
3) It is interesting that the results from the lineage tracing in adipose tissue in mice WT1-depemdent expressing CreER with tamoxifen induction at E14.5. The results show that showed that epididymal fat (77%), epicardial (66%), visceral (47%), and mesenteric (28%) Wt1+ cells exist in WAT tissue. What is the molecular mechanism underlying the WT1 promote adipose progenitor differentiation in adult adipose tissue not in embryo tissue?
Response:
Thank you for your comment. Recently, there was a negative suggestion of mesothelial cell involvement as a source of differentiation into adipocytes in visceral WAT. It was suggested that such involvement could lead to a deeper focus on the stromal vascular fraction in the future. Therefore, the significance of Wt1 in the stromal vascular fraction is currently unknown. Adipocytes with mesenchymal transition are known to be involved in the tumor microenvironmentWe have added the above information to the revised manuscript (page 7, lines 8–14) to implicate a link between the regulation of epithelial-mesenchymal transition possessed by Wt1 and adipocyte differentiation.
4) If WT1 is a potential therapeutic targets in regulating MC cells associated pathological diseases should be addressed in the discussion.
Response:
Thank you for your suggestion. We have added the medical significance of the functional regulation of mesothelial cells expressing Wt1 in the Conclusion section (Page 7, lines 11–12).

Reviewer 2 Report
Generally, The aspects of WT1 in regulating mesothelial cell Plasticity are interesting topics to be investigated in developmental and pathological processes, however, this review is too descriptive and lack the molecular mechanism in detail. The transgenic animal models or human genetic association relevances did not present in the review to justify how WT1 regulate the mesothelial cell plasticity during the developmental and pathological stages.
Specific points of views need to address:
1. The detail molecular mechanism underlying how does MS regulate surface or secreted the clycocalyx complex, such as HA, glycosaminoglycans and sialomucine proteins respond to extracellular matrix remodeling during the developmental and pathological processes.
2. The integrated logical informations linked to WT1 regulating chromosomal architecture, epigenomic control or promoters activity between fibroblast, myofibroblast and SMC differentiations respond to extra cellular matrix cue during the different developmental or pathological stages needed to be addressed in more detail.
3. It is interesting that the results from the lineage tracing in adipose tissue in mice WT1-depemdent expressing CreER with tamoxifen induction at E14.5. The results show that showed that epididymal fat (77%), epicardial (66%), visceral (47%), and mesenteric (28%) Wt1+ cells exist in WAT tissue. What is the molecular mechanism underlying the WT1 promote adipose progenitor differentiation in adult adipose tissue not in embryo tissue?
4. If WT1 is a potential therapeutic targets in regulating MC cells associated pathological diseases should be addressed in the discussion.
Author Response

(The authors gave the same response as above.)

Round 2
Reviewer 2 Report
The authors had done their best to address all the previous points and questions.